# Maternal Dietary Zinc Intake during Pregnancy and Childhood Allergic Diseases up to Four Years: The Japan Environment and Children’s Study

**DOI:** 10.3390/nu15112568

**Published:** 2023-05-30

**Authors:** Limin Yang, Miori Sato, Mayako Saito-Abe, Yumiko Miyaji, Mami Shimada, Chikako Sato, Minaho Nishizato, Natsuhiko Kumasaka, Hidetoshi Mezawa, Kiwako Yamamoto-Hanada, Yukihiro Ohya

**Affiliations:** Medical Support Center for the Japan Environment and Children’s Study, National Research Institute for Child Health and Development, Tokyo 157-8535, Japan

**Keywords:** zinc, allergy, maternal, children, cohort

## Abstract

Maternal dietary zinc intake and childhood allergy have inconsistent relationships. Thus, this study aimed to evaluate the influence of low maternal dietary zinc intake during pregnancy on developing pediatric allergic diseases. This study was designed using the Japan Environment and Children’s Study dataset. The model building used data from 74,948 mother–child pairs. Maternal dietary zinc intake was estimated based on the food frequency questionnaire, collecting the intake information of 171 food and beverage items. Fitted logistic regression models and generalized estimating equation models (GEEs) estimated the association between energy-adjusted zinc intake and childhood allergic conditions. The energy-adjusted zinc intake did not affect the risk of developing allergic disorders (wheeze, asthma, atopic dermatitis, rhinitis, and food allergy) in offspring. The GEE model revealed similar insignificant odds ratios. No significant association was found between zinc intake during pregnancy and allergic diseases in offspring in early childhood. Further study remains necessary to examine the association between zinc and allergy with reliable zinc status biomarkers in the body.

## 1. Introduction

Allergic diseases are major health issues in the young population, with the most common conditions including asthma, eczema/atopic dermatitis (AD), rhinitis, hay fever, and food allergies [1]. The prevalence of childhood allergic diseases is increasing in Japan and other developed countries over the past several decades [2]. Assumingly, the rapid increases in prevalence are associated with environmental exposure changes and childhood infection patterns. Additionally, these prevalence trends may be related to the altered population diet [3].

Zinc, an important element for humans, functions in catalyzing enzyme activity, thereby contributing to protein production, gene expression regulation, and cell membrane stabilization. Studies in the biochemistry and molecular biology fields revealed that over 300 enzymes and 2000 transcription factors require zinc to maintain their functions [4]. Additionally, zinc is a trace element with a poor plasma pool and rapid turnover. Maintaining a suitable dietary intake is necessary to ensure its steady state and maintain its functions due to the lack of storage in the body [5]. Two billion people suffer from zinc deficiency in developing countries, based on the World Health Organization estimation [4]. Zinc deficiency causes growth retardation, testicular hypofunction, increased oxidative stress, and increased pro-inflammatory cytokines generation [4]. Moreover, zinc is involved in anti-inflammatory processes as an antioxidant and anti-apoptotic agent and was found abundant in airway epithelial cells [6,7], suggesting a possible link between zinc and allergic disease development. Lower zinc levels in blood, nails, and hair were related to allergy development in children. However, the effect of maternal dietary zinc intake on allergy in offspring was weak, and the associations remain inconsistent, and some reports failed to show a significant association.

This study aimed to evaluate the influence of low maternal dietary zinc intake during pregnancy on the development of pediatric allergic diseases.

## 2. Materials and Methods

### 2.1. Study Design

The Japan Environment and Children’s Study (JECS) is an ongoing nationwide, multicenter, birth cohort study funded by the Ministry of the Environment, Japan, to clarify the relationship between environmental factors and children’s health and development [8]. A detailed study design description has been published elsewhere [8,9,10]. Briefly, the JECS is directly funded by the Ministry of the Environment, Japan, and recruited the participants from 15 regional centers (namely, Hokkaido, Miyagi, Fukushima, Chiba, Kanagawa, Koshin, Toyama, Aichi, Kyoto, Osaka, Hyogo, Tottori, Kochi, Fukuoka, and South Kyushu/Okinawa), which include both urban and rural areas in Japan [8,9]. The eligibility requirements for JECS participants were as follows: (1) those who resided in the study area at the time of recruitment and were expected to reside in Japan in the near future; (2) had an expected delivery date between 1 August 2011 and mid-2014; and (3) were capable of participating, i.e., they could understand the content of questionnaires and respond to them [8]. Moreover, JECS also recruited the participants contacted via cooperating healthcare providers or local government offices issuing the handbooks, and those who were willing to participate [8]. Self-administered questionnaires collecting information on the maternal history of diseases, dietary intake, and some baseline characteristics were given to the enrolled females in the first and second or third trimesters [10]. Follow-up surveys were conducted in their offspring until the child reached 13 years old. Maternal information on pregnancy, delivery, and newborns was transcribed by physicians, midwives/nurses, and/or research coordinators from medical records during early pregnancy, delivery, and 1 month after delivery [11].

This study was designed using the JECS dataset to analyze the association between maternal dietary zinc intake and allergic disease development in children aged 0–4 years. We used the dataset released in April 2021 (jecs-ta-20190930 and jecs-qa-20210401). Firstly, among the 104,059 records in the JECS-QA-20210401 dataset, 3759 miscarriages, stillbirths, or unknown birth outcomes were excluded. Thereafter, 1891 multiple births and 8 cases with unknown sex were removed. Among the remaining 98,391 records, 21,664 were not followed up after 4 years. Finally, those missing outcomes (1779 cases) were removed from the dataset, and 75,125 mother–child pairs were included for analysis (Figure 1).

The JECS protocol was reviewed and approved by the Ministry of the Environment’s Institutional Review Board on Epidemiological Studies and the Ethics Committees of all participating institutions (No. 100910001) [12,13,14]. Written informed consent was signed by all enrolled participants [12,13,14].

### 2.2. Maternal Dietary Zinc Intake

Maternal dietary zinc intake was estimated based on the food frequency questionnaire (FFQ) [15]. The FFQ used in JECS is a self-administered diet questionnaire. Validation for pregnant females was not performed, but it was validated with 12-day weighed food records in adults aged 40–74 years [15,16]. JECS performed the FFQ survey twice during pregnancy. The first survey was administered during the first trimester to investigate dietary patterns over the past year, while the second was conducted in the second to third trimesters to collect data on diet during pregnancy [17]. The latter was used for the current study. Pregnant females were asked about the frequency of consuming certain types of food or beverages, with 9 categories ranging from “never or less than once per month” to “at least 7 times per day”/(or “10 glasses per day for beverages”) [18]. Additionally, the quantity of consumed food or beverage was asked, with three portion size categories based on a given standard portion size. The daily intake of zinc was calculated using a food composition table developed from the Standardized Tables of Food Composition in Japan (2010 edition) [19]. We generated an energy-adjusted zinc intake by the residual method [17,20]. The energy-adjusted zinc intake was classified into quintiles. Q1 was the lowest quintile.

### 2.3. Allergic Diseases in Children

The primary outcome events include the development of allergic diseases in children aged 4 years. Outcomes variables in the models included “ever wheeze”, “current wheeze”, “ever asthma”, “ever atopic dermatitis (AD)”, “current AD,” ”ever rhinitis“, “current rhinitis”, “current food allergy”, and “any allergy”. Wheeze, asthma ever, AD, and rhinitis were evaluated using a questionnaire modified from the International Study of Asthma and Allergies in Childhood (ISAAC) for children aged 6–7 years, with translation validation in Japanese [21,22,23]. Additionally, JECS included questionnaires involving doctor-diagnosed allergic conditions. These are used to evaluate current food allergies (FAs). Moreover, “any allergy” was defined as suffering from any of the following conditions: current wheeze, rhinitis, AD, and FA. Outcome events (wheeze, AD, and FA) were repeatedly evaluated at 1~3 years old. These longitudinal data, in combination with the 4-year-old data, were used to reevaluate the association of maternal zinc intake and allergic diseases by fitting models, which explain repeated measurements. Detailed definitions are listed in Table 1.

### 2.4. Other Variables Used in the Models

Other variables included the sex of the child, parity, maternal age, pre-pregnancy overweight or obesity, socioeconomic state variables, smoking status, maternal drinking, feeding pet, history of maternal allergic diseases before pregnancy, premature birth, and low birth weight.

The data on maternal age, history of pregnancy and delivery, parity, pregnancy complications, gestational age, and birth weight were obtained from medical records and transcribed by research staff. Body mass index was calculated as weight divided by the square of height. A BMI value of ≥25 was defined as overweight or obese state of the mother. Premature birth was defined as gestational age of less than 37 weeks. Low birth weight was a birth weight of less than 2500 g. The detailed classification for these category variables is presented in Appendix A.

### 2.5. Statistical Methods

The logistic regression models were fitted to calculate the odds ratios (ORs) to evaluate the association between energy-adjusted dietary zinc intake and childhood allergic diseases. At first, sex, parity, maternal age, pre-pregnancy overweight or obesity, socioeconomic state variables, smoking status, maternal drinking, feeding pet, whether mother suffered allergic diseases before pregnancy, and energy-adjusted zinc intake during pregnancy were adjusted; thereafter, low birth weight and premature birth were further adjusted to check the change of coefficients. Confounding factors were selected according to the published association or clinical importance. The collinearity has been checked among independent variables, and a variance inflation factor of >5 was considered to have collinearity [24]. There was no multicollinearity among the variables mentioned above. Multiple imputations (MIs) were used to generate 10 datasets, of which missing data were imputed with a chained equation algorithm [24]. The pooled ORs from MI processes were compared with those from complete-case analysis (any missing variable was deleted).

Generalized estimating equation (GEE) models were developed with exchangeable working covariance for annually queried outcome events (current wheeze, AD, and FA) to explain repeated measurement.

In sensitivity analyses, in addition to the comparison of ORs from the abovementioned MI and complete-case analysis, models were fitted for energy-adjusted maternal zinc intake, which was treated as a continuous variable. A nonlinear part was generated for the energy-adjusted maternal dietary zinc intake in these models using restricted cubic splines. To evaluate the interaction effect, an interaction term (maternal history of allergic diseases × energy-adjusted zinc intake) was added into the models. Moreover, models were built by adding variable “allergic disease history of father” into the model to adjust the effect of the paternal factor on the child’s allergy. Because the missing rate of this variable was around 50%, non-respondents were excluded. Finally, to exclude the effect from taking supplements during pregnancy on outcomes, all the models were refitted in a subgroup, which excluded those who took supplements during pregnancy. All the analyses were performed using R software, version 4.2.3 (Institute for Statistics and Mathematics, Vienna, Austria; www.r-project.org accessed on 1 May 2023).

## 3. Results

### 3.1. Baseline Characteristics of the Participants

Baseline characteristics are presented in Appendix A. Among the 74,948 mothers eligible for analysis, the prevalence rates of current wheeze, AD, rhinitis, and caregiver-reported doctor-diagnosed FA were 14.2%, 13.5%, 31.5%, and 5.6%, respectively. Low family income was observed in 38.4% of the study population. Boys account for 51.2% of the children. Around 58.3% of pregnant women reported a history of allergic diseases. In pregnant women, the smoking and drinking rates were 3.6% and 2.7%, respectively. Smoking was observed in 44.3% of men in our study population. Around 7.9% of the infants were born with a weight less than 2500 g, and approximately 4.5% of deliveries occurred earlier than 37 weeks.

Baseline characteristics were compared with those excluded from the analysis due to the sample selection process (Appendix A). Slight differences were found between the selected and the excluded cases. The sample cohort used for analysis revealed the following characteristics compared to those excluded: higher socioeconomic status and education levels; lower rates of smoking, drinking, maternal obesity, preterm birth, and low birth weight of children.

### 3.2. Association between Maternal Dietary Zinc during Pregnancy and Children’s Allergic Diseases

Table 2 shows the logistic models that estimate the effect of maternal dietary zinc intake on the development of allergy at 4 years. The ORs revealed that energy-adjusted zinc intake during pregnancy did not affect the risk of developing allergic disorders in offspring at 4 years old, after adjusting confounders. Further adjusting birth weight and premature birth did not change the ORs.

GEE models were fitted to explore the association between maternal dietary zinc intake with childhood allergy for the longitudinal data, and the results from the models are listed in Table 3. We did not find any significant ORs suggesting an association of energy-adjusted zinc intake during pregnancy with the onset of wheeze, AD, or FA during the 0–4-year-old period, other than a slightly increased OR for FA observed in the Q1 level, even after interpreting repeated measurements. This slightly increased OR was obtained by chance, without implying a meaningful association, because the OR is very close to 1, and this analysis has multiple statistical tests.

### 3.3. Sensitivity Analysis

The association of maternal dietary zinc with allergic diseases in offspring was re-evaluated with a dataset excluding any data with missing values (Appendix A). The ORs from the complete cases were similar to those obtained from the MI processes. No significant coefficient was found in the fitted models when energy-adjusted zinc intake was fitted as a continuous variable (Appendix A). Additionally, all the nonlinear parts were insignificant (Appendix A). The interaction effect between energy-adjusted zinc intake and maternal history of allergy was checked. The Wald statistics list is shown in Appendix A. No significant interaction effect was observed. Models that further adjusted paternal history of the allergic disease were fitted (Appendix A). No significant increased odds of developing allergic diseases were observed in low or high maternal dietary zinc intake levels, compared with normal dietary intake. Logistic models in subgroups excluding those who took vitamins or supplements during pregnancy are listed in Appendix A. No significant changed result was observed, compared with the ORs shown in Table 2.

## 4. Discussion

This study investigated the effects of maternal dietary zinc intake in the development of allergic diseases in offspring with a large longitudinal dataset. No associations were found between energy-adjusted zinc intake during pregnancy and allergic diseases in offspring after adjusting the confounders.

Several studies investigated the correlation between zinc levels (using serum and erythrocyte zinc measurements, hair/nails zinc, supplementation, and dietary zinc intake) and the development of allergic diseases. No significant association was found between maternal zinc intake during pregnancy and allergic diseases in offspring, which is consistent with some of the previous studies. Results from a birth cohort study, including 450 mother–infant pairs, suggest that sufficient maternal dietary intakes have no protective effects on the risk of childhood allergic diseases [25]. Another birth cohort study that was conducted in Japan with 763 cases indicated no statistical association between maternal zinc intake and the risk of wheezing or eczema in the offspring aged 16–24 months [26]. A study in the United Kingdom based on 1924 children revealed no associations between maternal intake of zinc and wheezing or eczema in the first 2 years of life [3]. Additionally, studies focused on umbilical cord trace elements do not seem to add evidence to these associations. The Avon Longitudinal Study of Parents and Children measured zinc levels in the umbilical cord and indicated that umbilical cord zinc levels did not predict the development of pediatric allergic diseases [27]. However, other studies revealed a beneficial correlation. For example, a cohort study with 1290 mother–child pairs revealed significant protection of maternal zinc intake for any wheezing in the highest quartile compared with the lowest quartile of intake [6]. However, the study conducted multiple statistical tests, thus it cannot be ruled out that the significant *p*-value was obtained only by chance. Beckhaus et al. performed a meta-analysis to summarize maternal nutrition during pregnancy and the risk of asthma, wheezing, and ADs during childhood. They demonstrated that high maternal zinc intake decreased the risk of developing childhood wheezing. However, a similar beneficial effect has not been found in asthma, eczema, allergic rhinitis, FA, and STR to allergens [28]. These conflicting results are postulated to be due to the differences in sample size and study design, and a lack of consistent variable definitions.

Zinc is one of the trace elements that has crucial functions in antioxidant defense, DNA repair, and modulating immunity [6]. Moreover, zinc is involved in the differentiation and production of T helper cells [4]. The study also suggested that zinc is required in the process of mRNA generation of cytokines (interferon-γ and interleukin [IL]-2) secreted from Th1 cells, and zinc inhibits NF-kB activation, resulting in decreased cytokine and molecular expression (tumor necrosis factor-a, IL-1b, and VCAM) [4]. Similarly, zinc deficiency causes reduced Th1 cytokine secretion and enhanced Th2 cytokine responses [27]. Additionally, zinc affects some important proteins related to the airway, such as ADAM33 metalloproteinase, and β2 adrenoreceptors [26,29]. Therefore, inadequate maternal zinc intake may affect fetal airway development. An animal study demonstrated that zinc deficiency was associated with abnormal prenatal lung growth in rats [30]. However, our findings did not support the association of zinc intake during pregnancy with the development of allergic diseases in the offspring. We acknowledge that this result might have been obtained because zinc intake during pregnancy is an inaccurate proxy indicator of the exact level of zinc in the fetus. Further research is still required to determine whether the association of zinc with allergy occurs in the fetus and affects the development of the disease after birth.

This study had some advantages. First, the sample size is large, which ensures sufficient power to detect weak, but significant associations and allows more variable adjustment in the models [24]. Additionally, the longitudinal study design may reduce the chances of recall bias.

The following limitations of the study should be acknowledged. First JECS is not a population-based study, although the sample size is approximately 100,000, and the demographic characteristic is similar to the Japanese population [31]. JECS revealed lost follow-up regarding all large birth cohorts. Dropout and missing follow-up were more likely to occur in mothers with low-income households, parents with low educational attainment, and smoking parents [31]. However, these are not thought to introduce significant bias into our results, because these discrepancies were not so severe. Second, some discrepancies remained between FFQ-derived estimates and actual maternal dietary nutrient intake. FFQ-derived estimates are influenced by the subjective recall of foods consumed and their quantities [3], and the real intake is affected by body absorption and metabolic ability [32]. Moreover, maternal dietary intake is a poor proxy that reflects the zinc status in the fetus, because the concentrates of zinc in the fetus are not only related to the mother’s dietary intake, but are also influenced by the mother’s absorption efficiency, the amount that passes through the placenta, and the fetus’s demand [33]. The misclassification of exposure is likely to be non-differential and tends to attenuate associations [3], thus the ORs presented in this study might underestimate the true associations. Further study using a better proxy of zinc concentrations remains necessary. Third, the presence of allergic diseases was defined according to the caregiver’s report, which may have resulted in some misclassification for allergic diseases [17]. However, the questionnaires based on the ISAAC format were extensively used in epidemiological studies and were shown to be reproducible [3,34,35,36]; hence, inaccurate estimates of outcome events due to the ISAAC questionnaire will cause a large impact on the study results. These non-differential misclassifications might only bring a slight underestimation of the associations. Finally, evaluating the effect of specific nutrients perhaps oversimplifies the problem, as other similar observational studies have pointed out. Excluding the effects of some co-intake of other protective nutrients is difficult [25]. These variables cannot be adjusted in the model as confounders due to the multicollinearity problems.

## 5. Conclusions

This study revealed no significant effects of maternal zinc intake on any allergic diseases in children. Therefore, further studies remain necessary to examine the association between zinc and allergy, with reliable biomarkers for zinc status in the body.

## Figures and Tables

**Figure 1 nutrients-15-02568-f001:**
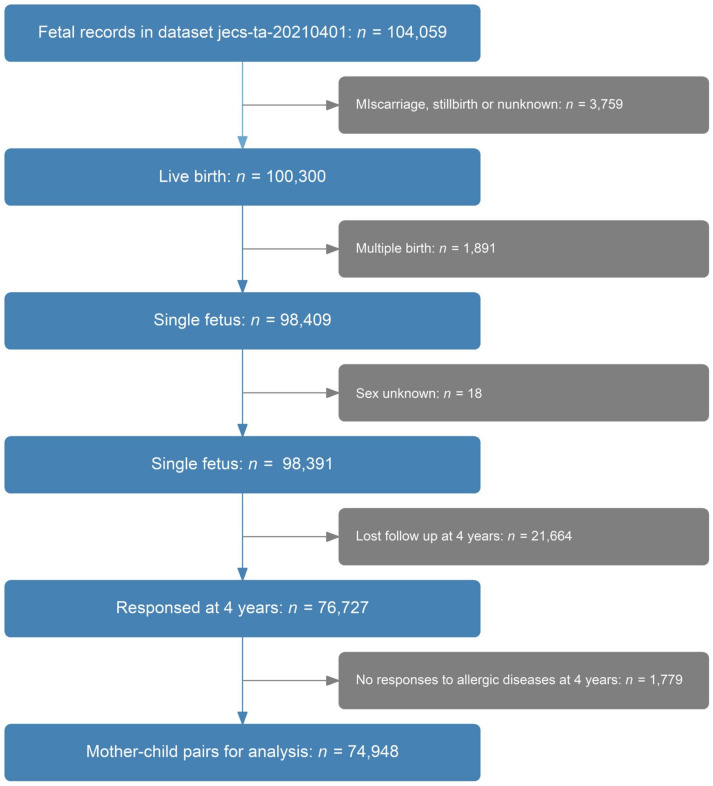
Sample cohort selection.

**Table 1 nutrients-15-02568-t001:** Definitions of outcome events.

Outcome Events	Question	Answer
Current wheeze at 4 years	“Has your child ever had wheezing or whistling in the past 12 months?”	Yes
Ever wheeze at 4 years	“Has your child ever had wheezing or whistling in the chest at any time in the past?”	Yes
Ever asthma at 4 years	“Has your children ever been diagnosed as asthma by a doctor?”	Yes
Ever rhinitis at 4 years	“Has your child ever had a problem with sneezing, or a runny, or blocked nose when he/she DID NOT have a cold or the flu?”	Yes
Rhinitis at 4 years	“In the past 12 months, has your child had a problem with sneezing, or a runny or blocked nose when he/she did not have a cold or the flu?”	Yes
Ever AD at 4 years	“Has your child ever had atopic dermatitis?”	Yes
Current AD at 4 years	(1) “Has your child had itchy rash at any time in the past 12 months?” (2) “Has this itchy rash at any time affected any of the following places: the folds of the elbows, behind the knees, in front of the ankles, under the buttocks, or around the neck, ears or eyes?”	Yes
Current FA at 4 years	“Has your child ever been diagnosed by a physician as having food allergy in the past 6 months?”	Yes

AD: atopic dermatitis; FA: food allergy.

**Table 2 nutrients-15-02568-t002:** Odds ratios from logistic regression models with multiple imputation dataset.

			95% CI			95% CI	
		OR ^#^	Lower	Upper	OR ^&^	Lower	Upper
Ever wheeze	Q1	1.001	0.952	1.053	1.001	0.952	1.053
	Q2	0.972	0.924	1.022	0.971	0.923	1.022
	Q3	1.000			1.000		
	Q4	0.962	0.915	1.013	0.961	0.914	1.011
	Q5	0.994	0.945	1.045	0.991	0.942	1.043
Current wheeze	Q1	0.967	0.906	1.032	0.967	0.906	1.032
	Q2	0.952	0.892	1.016	0.951	0.891	1.015
	Q3	1.000			1.000		
	Q4	0.914	0.855	0.976	0.912	0.854	0.974
	Q5	1.001	0.939	1.069	0.999	0.936	1.066
Ever asthma	Q1	1.021	0.951	1.095	1.021	0.951	1.095
	Q2	0.971	0.904	1.042	0.970	0.903	1.042
	Q3	1.000			1.000		
	Q4	0.950	0.884	1.021	0.949	0.883	1.020
	Q5	1.057	0.985	1.134	1.055	0.983	1.132
Ever AD	Q1	1.044	0.974	1.118	1.043	0.974	1.118
	Q2	0.989	0.923	1.060	0.989	0.923	1.060
	Q3	1.000			1.000		
	Q4	0.931	0.868	0.998	0.931	0.868	0.999
	Q5	1.002	0.935	1.075	1.003	0.936	1.075
Current AD	Q1	1.013	0.948	1.082	1.013	0.948	1.082
	Q2	1.018	0.953	1.088	1.018	0.953	1.088
	Q3	1.000			1.000		
	Q4	0.954	0.892	1.020	0.955	0.893	1.021
	Q5	0.961	0.899	1.028	0.962	0.900	1.029
Ever rhinitis	Q1	1.046	0.996	1.097	1.045	0.996	1.097
	Q2	0.972	0.927	1.020	0.972	0.927	1.020
	Q3	1.000			1.000		
	Q4	0.979	0.933	1.027	0.979	0.933	1.028
	Q5	1.010	0.962	1.060	1.010	0.963	1.060
Current rhinitis	Q1	1.035	0.985	1.087	1.035	0.985	1.087
	Q2	0.978	0.931	1.027	0.978	0.931	1.027
	Q3	1.000			1.000		
	Q4	0.977	0.930	1.026	0.977	0.930	1.027
	Q5	0.994	0.946	1.044	0.994	0.946	1.044
Current FA	Q1	1.059	0.958	1.169	1.058	0.958	1.169
	Q2	1.012	0.915	1.119	1.012	0.915	1.119
	Q3	1.000			1.000		
	Q4	0.987	0.892	1.091	0.988	0.893	1.092
	Q5	1.069	0.968	1.180	1.070	0.969	1.181
Any allergy	Q1	1.024	0.977	1.072	1.024	0.977	1.072
	Q2	0.982	0.938	1.028	0.981	0.937	1.028
	Q3	1.000			1.000		
	Q4	0.953	0.911	0.998	0.954	0.911	0.998
	Q5	0.973	0.929	1.019	0.973	0.929	1.019

AD: atopic dermatitis; FA: food allergy; ORs: odds ratios; CI: confidence interval. ^#^ The models adjusted sex, parity, maternal age, pre-pregnancy overweight or obesity, socioeconomic indicators, smoking status, maternal drinking, feeding pet, maternal history of allergic diseases before pregnancy, and energy-adjusted zinc intake during pregnancy. **^&^** The models further adjusted low birth weight and premature birth.

**Table 3 nutrients-15-02568-t003:** Odds ratios from generalized estimating equation models.

			95% CI			95% CI	
		OR ^#^	Lower	Upper	OR ^&^	Lower	Upper
Current wheeze	Q1	1.027	0.986	1.071	1.028	0.986	1.071
	Q2	0.969	0.929	1.010	0.968	0.928	1.009
	Q3	1.000			1.000		
	Q4	0.981	0.941	1.023	0.980	0.940	1.022
	Q5	1.013	0.972	1.056	1.011	0.970	1.054
Current AD	Q1	1.014	0.965	1.067	1.014	0.965	1.066
	Q2	0.991	0.942	1.042	0.991	0.942	1.042
	Q3	1.000			1.000		
	Q4	0.921	0.875	0.969	0.921	0.876	0.969
	Q5	0.943	0.896	0.992	0.944	0.897	0.994
Current FA	Q1	1.080	1.007	1.158	1.080	1.007	1.158
	Q2	1.051	0.980	1.127	1.051	0.980	1.127
	Q3	1.000			1.000		
	Q4	0.968	0.902	1.039	0.969	0.903	1.041
	Q5	0.971	0.904	1.042	0.972	0.906	1.043

AD: atopic dermatitis; FA: food allergy; ORs: odds ratios; CI: confidence interval. ^#^ The models adjusted sex, parity, maternal age, pre-pregnancy overweight or obesity, socioeconomic state variables, smoking status, maternal drinking, feeding pet, maternal history of allergic diseases before pregnancy, and energy-adjusted zinc intake during pregnancy. ^&^ The models further adjusted low birth weight and premature birth.

## Data Availability

The data are unsuitable for public deposition due to the ethical restrictions and legal framework of Japan. It is prohibited by the Act on the Protection of Personal Information (Act No. 57 of 30 May 2003, amendment on 9 September 2015) to publicly deposit data containing personal information. Ethical Guidelines for Medical and Health Research Involving Human Subjects enforced by the Japan Ministry of Education, Culture, Sports, Science and Technology and the Ministry of Health, Labour and Welfare also restrict the open sharing of the epidemiologic data. All inquiries about access to data should be sent to: jecs-en@nies.go.jp. The person responsible for handling enquiries sent to this e-mail address is Dr Shoji F. Nakayama, JECS Programme Office, National Institute for Environmental Studies.

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
