# Peer review of "Maternal Dietary Zinc Intake during Pregnancy and Childhood Allergic Diseases up to Four Years: The Japan Environment and Children’s Study"

_nutrients, 2023, doi:10.3390/nu15112568_

Round 1

Reviewer 1 Report

This manuscript investigated the relationship between maternal zinc intake during pregnancy and childhood allergic diseases, but there are still some issues with the design of this work. The following are comments on this manuscript.

1. The author could consider whether the zinc could be metabolized after maternal intake dietary zinc? what is the eventual concentration of zinc in mothers? And whether the zinc will transfer to children?

2. The author should improve the mechanism description about the relationship between maternal dietary zinc intake during pregnancy and childhood allergic diseases.

3. Why the author overlooked the impact of dietary supplements? Did the questionnaires ask any question about supplement intake? If not, this study might be not accurate, since the supplement might significantly alter the zinc levels in the body.

4. In this study, the calculation of dietary zinc intake mainly refers to the “Standardized Tables of Food Composition in Japan”, which is not mentioned in the appendix. Moreover, this standard was issued ten years ago, the dietary structure of food may be undergone significant changes within the recent ten years. It is worth considering whether this calculation method is still suitable.

5. In this study, the author designs the scheme mainly considered the mother's factors. Actually, the father's factors, family relationships, environmental and other complicating factors should also be considered.

6. The result would be more interesting if the authors investigate the correlation between the zinc levels in the mother and maternal dietary zinc intake, and then evaluate the effect of maternal dietary zinc intake on changing the zinc levels thereby affecting the development of pediatric allergic diseases. 

Author Response

Reviewer 1

This manuscript investigated the relationship between maternal zinc intake during pregnancy and childhood allergic diseases, but there are still some issues with the design of this work. The following are comments on this manuscript.

  1. The author could consider whether the zinc could be metabolized after maternal intake dietary zinc? what is the eventual concentration of zinc in mothers? And whether the zinc will transfer to children?

A: Thank you for the comment. Unfortunately, data regarding the concentration of maternal/fetal zinc in plasma or hair were unavailable for analysis in this study. Moreover, it is beyond the scope of this study to determine whether there is a relationship between nutritional intake of zinc and maternal/fetal zinc concentration. This study used maternal dietary intake as a proxy indicator to reflect maternal zinc levels. We acknowledge that this is not an accurate proxy indicator. Therefore, we have mentioned it as a limitation of the study in the Discussion section.

  1. The author should improve the mechanism description about the relationship between maternal dietary zinc intake during pregnancy and childhood allergic diseases.

A: Thank you for the comment. We have revised this part as follows:

Zinc is one of the trace elements that having crucial functions in antioxidant defense, DNA repair, and modulating immunity. Moreover, zinc is involved in the differenti-ation and production of T helper cells. The study also suggested that zinc is required in the process of mRNA generation of cytokines (interferon-γ and interleukin [IL]-2) secreted from Th1 cells, and zinc inhibits NF-kB activation, resulting in decreased cytokine and molecular expression (tumor necrosis factor-a, IL-1b, and VCAM). Similarly, zinc deficiency causes reduced Th1 cytokine secretion and enhanced Th2 cytokine responses. Additionally, zinc affects some important proteins related to the airway, such as ADAM33 metalloproteinase and β2 adrenoreceptors. Therefore, inadequate ma-ternal zinc intake may affect fetal airway development. An animal study demonstrated that zinc deficiency was associated with abnormal prenatal lung growth in rats. However, our findings did not support the association of zinc intake during pregnancy with the development of allergic diseases in the offspring. We acknowledge that this result might be obtained because zinc intake during pregnancy is an inaccurate proxy indicator of the exact level of zinc in the fetus. Further research is still required to determine whether the association of zine with allergy occurs in the fetus and affects the development of the disease after birth.

  1. Why the author overlooked the impact of dietary supplements? Did the questionnaires ask any question about supplement intake? If not, this study might be not accurate, since the supplement might significantly alter the zinc levels in the body.

A: Thank you for the comment. Because there were no data on zinc supplements in JECS, we could not adjust for this variable in the model. However, we have added a subgroup analysis in the revised manuscript to exclude the effects of zinc supplements. In this subgroup analysis, we have refit the models with data of those individuals who had not taken the vitamin or supplements during pregnancy.

  1. In this study, the calculation of dietary zinc intake mainly refers to the “Standardized Tables of Food Composition in Japan”, which is not mentioned in the appendix. Moreover, this standard was issued ten years ago, the dietary structure of food may be undergone significant changes within the recent ten years. It is worth considering whether this calculation method is still suitable.

A: Thank you for your comment. We have used the FFQ to calculate zinc intake during pregnancy, which is widely used in nutrition research. Moreover, the use of FFQ by the JECS was evaluated in 2016, and the validity has been proven.

  1. In this study, the author designs the scheme mainly considered the mother's factors. Actually, the father's factors, family relationships, environmental and other complicating factors should also be considered.

A: Thank you for your suggestion. Accordingly, we have added the required information (Table S7).

  1. The result would be more interesting if the authors investigate the correlation between the zinc levels in the mother and maternal dietary zinc intake, and then evaluate the effect of maternal dietary zinc intake on changing the zinc levels thereby affecting the development of pediatric allergic diseases. 

A: Thank you for the comment. We could not perform a mediation analysis to evaluate the mediation effect of maternal or fetal zinc levels despite of a rational study design because we did not have data on maternal or fetal zinc levels.

Reviewer 2 Report

It is opinion of the reviewer, that this paper before acceptance needs several corrections. My individual comments are listed below.

The title should be written with capital letters.

P. 1 – The authors’ e-mail addresses and initialts should be completed.

Abstract – The allergic diseases should be mentioned.

P. 1 – The presence of similar research in countries should be described.

P. 1 – It should be “”… in biochemistry and molecular biology field …”.

This research should be compared in few words with previous Japanese research.

P. 2 – It should be “pro-inflammatory cytokines ….”.

P. 2 – It should be “… in blood, nails …”.

P. 3 – “OR#” and “OR&” needs to be explained.

P; 4 – The footnote – It should be “… or obesity”.

P. 5, 2.5 - The description should be impersonal (for example “The logistic regression model was used to …”.

P. 5 The description should be impersonal “3.1  baseline characteristics …”.

P. 7 – Zinc cannot be called as “antioxidant.

P. 8, Conclusions  – What does it mean “… nail micronutrients”?

Reference#14 – It should be “PLoS ONE”.

Reference #24 – The journal title abbreviation is needed.

It is opinion of the reviewer, that this paper before acceptance needs several corrections. My individual comments are listed below.

The title should be written with capital letters.

P. 1 – The authors’ e-mail addresses and initialts should be completed.

Abstract – The allergic diseases should be mentioned.

P. 1 – The presence of similar research in countries should be described.

P. 1 – It should be “”… in biochemistry and molecular biology field …”.

This research should be compared in few words with previous Japanese research.

P. 2 – It should be “pro-inflammatory cytokines ….”.

P. 2 – It should be “… in blood, nails …”.

P. 3 – “OR#” and “OR&” needs to be explained.

P; 4 – The footnote – It should be “… or obesity”.

P. 5, 2.5 - The description should be impersonal (for example “The logistic regression model was used to …”.

P. 5 The description should be impersonal “3.1  baseline characteristics …”.

P. 7 – Zinc cannot be called as “antioxidant.

P. 8, Conclusions  – What does it mean “… nail micronutrients”?

Reference#14 – It should be “PLoS ONE”.

Reference #24 – The journal title abbreviation is needed.

Author Response

Reviewer 2

It is opinion of the reviewer, that this paper before acceptance needs several corrections. My individual comments are listed below.

The title should be written with capital letters.

A: Thank you for the comment. Accordingly, we have revised the title.

  1. 1 – The authors’ e-mail addresses and initialts should be completed.

A: Thank you for the comment. The E-mail of the corresponding author has been added.

Abstract – The allergic diseases should be mentioned.

A: Thank you for the comment. The sentence has been revised as follows:

The energy-adjusted zinc intake did not affect the risk of developing allergic disorders (wheeze, asthma, atopic dermatitis, rhinitis, and food allergy) in their offspring.

  1. 1 – The presence of similar research in countries should be described.

A: Thank you for the comment. Similar studies are listed in the Discussion section.

  1. 1 – It should be “”… in biochemistry and molecular biology field …”.

A: Thank you for the comment. Accordingly, we have revised the sentence.

This research should be compared in few words with previous Japanese research.

A: Thank you for the comment. Accordingly, we have compared our study with a Japanese research in the Discussion section as shown below:

Another birth cohort study that was conducted in Japan with 763 cases indicated no statistical association between maternal zinc intake and the risk of wheezing or eczema in the offspring aged 16–24 months.

  1. 2 – It should be “pro-inflammatory cytokines ….”.

A: Thank you for the comment. The sentence has been revised.

  1. 2 – It should be “… in blood, nails …”.

A: Thank you for the comment. The sentence has been revised.

  1. 3 – “OR#” and “OR&” needs to be explained.

A: Thank you for the comment. “OR#” and “OR&” have been explained in the footnote of Table 2

P; 4 – The footnote – It should be “… or obesity”.

A: Thank you for the comment. The word has been revised.

  1. 5, 2.5 - The description should be impersonal(for example “The logistic regression model was used to …”.

A: Thank you for the comment. The sentence has been revised.

  1. 5 The description should be impersonal“3.1  baseline characteristics …”.

A: Thank you for the comment. The sentence has been revised.

  1. 7 – Zinc cannot be called as “antioxidant.

A: Thank you for the comment. The sentence has been revised.

  1. 8, Conclusions  – What does it mean “… nail micronutrients”?

A: Thank you for the comment. The sentence has been revised.

Reference#14 – It should be “PLoS ONE”.

A: Thank you for the comment. Reference 14 has been revised.

Reference #24 – The journal title abbreviation is needed

A: Thank you for the comment. Reference 24 has been revised.